# ATF3/SPI1/SLC31A1 Signaling Promotes Cuproptosis Induced by Advanced Glycosylation End Products in Diabetic Myocardial Injury

**DOI:** 10.3390/ijms24021667

**Published:** 2023-01-14

**Authors:** Shengqi Huo, Qian Wang, Wei Shi, Lulu Peng, Yue Jiang, Mengying Zhu, Junyi Guo, Dewei Peng, Moran Wang, Lintong Men, Bingyu Huang, Jiagao Lv, Li Lin

**Affiliations:** Division of Cardiology, Department of Internal Medicine, Tongji Hospital, Tongji Medical College, Huazhong University of Science and Technology, Wuhan 430030, China

**Keywords:** cuproptosis, advanced glycosylation end products, diabetic cardiomyopathy, SLC31A1

## Abstract

Cuproptosis resulting from copper (Cu) overload has not yet been investigated in diabetic cardiomyopathy (DCM). Advanced glycosylation end products (AGEs) induced by persistent hyperglycemia play an essential role in cardiotoxicity. To clarify whether cuproptosis was involved in AGEs-induced cardiotoxicity, we analyzed the toxicity of AGEs and copper in AC16 cardiomyocytes and in STZ-induced or db/db-diabetic mouse models. The results showed that copper ionophore elesclomol induced cuproptosis in cardiomyocytes. It was only rescued by copper chelator tetrathiomolybdate rather than by other cell death inhibitors. Intriguingly, AGEs triggered cardiomyocyte death and aggravated it when incubated with CuCl_2_ or elesclomol–CuCl2. Moreover, AGEs increased intracellular copper accumulation and exhibited features of cuproptosis, including loss of Fe–S cluster proteins (FDX1, LIAS, NDUFS8 and ACO2) and decreased lipoylation of DLAT and DLST. These effects were accompanied by decreased mitochondrial oxidative respiration, including downregulated mitochondrial respiratory chain complex, decreased ATP production and suppressed mitochondrial complex I and III activity. Additionally, AGEs promoted the upregulation of copper importer SLC31A1. We predicted that ATF3 and/or SPI1 might be transcriptional factors of SLC31A1 by online databases and validated that by ATF3/SPI1 overexpression. In diabetic mice, copper and AGEs increases in the blood and heart were observed and accompanied by cardiac dysfunction. The protein and mRNA profile changes in diabetic hearts were consistent with cuproptosis. Our findings showed, for the first time, that excessive AGEs and copper in diabetes upregulated ATF3/SPI1/SLC31A1 signaling, thereby disturbing copper homeostasis and promoting cuproptosis. Collectively, the novel mechanism might be an alternative potential therapeutic target for DCM.

## 1. Introduction

The acquisition, distribution and elimination of metal micronutrients, such as iron (Fe) and copper (Cu), are essential for life [1]. Metabolic diseases such as diabetes, obesity and non-alcoholic fatty liver usually involve the interaction of multiple underlying molecular mechanisms [2,3,4]. Multiple lines of evidence have shown that metabolic diseases including obesity and diabetes are usually accompanied by the dysfunctional regulation of various metal ions [5,6]. Among them, copper is central to many important biological processes, including mitochondrial respiration, antioxidant defense and biocompound synthesis [1,7], and is particularly closely associated with the severity and progression of diabetes [8]. Serum copper was significantly increased in diabetic patients and an STZ-induced diabetic rat model [8,9,10]. Experiments in vitro and in vitro showed that coronary perfusion with a low concentration of copper ions could significantly reduce cardiac function [11], and the bivalent copper chelator triethylenetetramine could significantly improve cardiac function in rats with diabetic cardiomyopathy [12,13]. This experimental evidence indicated that copper homeostasis might play an important role in maintaining cardiac function. However, how copper ions influence cardiac function in diabetic cardiomyopathy remains elusive [10,14,15].

Recently, a unique insight into programmed cell death, referred to as cuproptosis, was proposed in tumor cells and ATP7B^−/−^ mice [16]. This copper-induced cell death is characterized by, on the one hand, the decrease in lipoylation of DLAT and DLST and the oligomerization of lipoylated proteins of the TCA cycle induced by the direct copper binding and, on the other hand, destabilization and overall reduction in iron–sulfur (Fe–S) cluster proteins, which together lead to proteotoxic stress and mitochondrial dysfunction [16]. The Fe–S cluster protein FDX1 and lipoylation are key regulators of cuproptosis [16]. However, cuproptosis in cardiomyocytes has not been reported, and whether cuproptosis is involved in diabetic cardiomyopathy is still undetermined.

Cu homeostasis is tightly maintained in all organisms through mechanisms of uptake, transport, storage and excretion at a precise scale [1]. The upper limit of loosely bound copper is restricted to an almost vanishingly low level of less than a single atom per cell [17]. Redox Cu(II) bound to plasma protein carriers (ceruloplasmin) is predominant in the blood. Cu(II) is reduced to Cu(I) by reductases and transported into the cytoplasm by the high-affinity importer SLC31A1 (CTR1). The imported Cu(I) binds metallothioneins (GSH, etc.) and metallochaperones (superoxide dismutase, etc.), which distribute Cu(I) to different subcellular locations [1,18]. Copper ionophores such as elesclomol are copper-binding small molecules that shuttle copper into the cell, ignoring the ion concentration gradient. Therefore, they are useful tools to study copper toxicity [7,19]. A series of studies have indicated that the mechanism of copper ionophore-induced cell death involves intracellular copper accumulation and not the effect of the small molecule chaperones themselves [16].

Advanced glycosylation end products (AGEs) are a heterogeneous group of compounds that include more than 20 different products which are produced by glycation, called the Maillard reaction [20]. Glucose, fructose or more reactive dicarbonyls react nonenzymatically with nucleophilic groups on proteins, preferentially with ε-amino groups of lysines and N-terminal-amino groups, following a succession of rearrangements of intermediate compounds and subsequently converting to stable AGEs [20,21]. AGEs react with new proteins, perpetuating and propagating oxidative modifications and producing new AGEs crosslinks and then accumulating gradually [21]. Although this reaction is very slow under physiological conditions, this protein modification by glucose is significantly accelerated under the hyperglycemic conditions of diabetes and is considered one of the major factors in the development of diabetic complications [21,22,23]. More importantly, oxidative reactions catalyzed by Cu^2+^ and Fe^3+^ redox metal ions are involved in AGEs formation under in vivo conditions, but these metal ions are sequestered in specific metal transporters and other metalloproteins in blood plasma or the cellular cytoplasm [21]. Copper metalloproteins after glycation reactions undergo significant conformational changes and even fragmentation, causing the release of bound metal, completing a positive-feedback loop [24,25]. AGEs could promote cardiomyocyte death through calcium overload, oxidative stress or excessive autophagy [26,27,28]. However, it is unclear whether cuproptosis is involved in AGEs-induced cardiomyocyte death. Thus, we investigated the potential effect of cuproptosis on AGEs-induced cardiomyocyte dysfunction in diabetic cardiomyopathy.

## 2. Results

### 2.1. Copper Ionophores Induced Cuproptosis, a Distinct Form of Regulated Cell Death, in Cardiomyocytes

We screened the cytotoxic effects of multiple structurally distinct copper ionophores in cardiomyocytes and found that elesclomol and NSC-319726 combined with copper chloride had significant cytotoxic effects on AC16 cardiomyocytes, whereas the copper ionophore-induced cell death was mild (Figure 1A and Appendix A). To further confirm that copper ionophore cytotoxicity is selectively dependent on copper, we analyzed the killing potential of the potent copper ionophore elesclomol combined with various metal ion chlorides. Only copper aggravated the cytotoxic effect in cardiomyocytes, but supplementation with other metals, including iron, cobalt, zinc, magnesium and nickel, failed to potentiate cell death more than elesclomol treatment alone (Figure 1B). This illustrated that elesclomol had the highest selectivity for copper ions. Two-hour pulse treatment with elesclomol–copper chloride triggered cell death significantly (Figure 1C). Meanwhile, the cells were treated with two-hour pulse elesclomol–copper chloride (ES–Cu) and then washed with new medium to remove the ES–Cu. These cells were cultured for an additional 24 h or 48 h, and the CCK8 assay revealed that the cell viability decreased with continued culture (Figure 1C). This result suggests that short-term exposure to copper ionophores with copper leads to irrevocable, subsequent cell cytotoxicity.

We used caspase 3/7 positive cells to evaluate the apoptotic cell level. The caspase 3/7 positive cells were green fluorescent. The number of total cells in the white, bright image was calculated by Image J software. The quantification of the caspase 3/7 activity was based on the number of green fluorescent positive cells compared to the total cell numbers. Bortezomib could target multiple pathways including p53, the nuclear factor kappa B, the phosphatidylinositol 3 kinase pathway and the ubiquitin/proteasome pathway to promote apoptosis [29]. The Figure 1D revealed that bortezomib induced a 20- to 30-fold green–positive apoptotic cell increase. However, the number of green–positive apoptotic cells after elesclomol–CuCl_2_ treatment did not increase. Elesclomol–CuCl_2_ induced cell death without caspase 3/7 cleavage, which is significantly different from the cell death induced by bortezomib. Meanwhile, cleaved caspase 3 and cleaved PARP were elevated after treatment with the apoptosis inducers bortezomib and etoposide, but not after elesclomol–CuCl2 treatment (Figure 1E). Similarly, elesclomol–CuCl2 killing potential was maintained when cardiomyocytes were cotreated with pancaspase inhibitors (Z-VAD-FMK and Boc-D-FMK) (Figure 1F), again indicating that the copper-induced cell death is distinct from apoptosis. Furthermore, treatment with inhibitors of other known cell death mechanisms, including ferroptosis (ferrostatin-1), necroptosis (necrostatin-1) and oxidative stress (N-acetyl cysteine), failed to abrogate copper-induced cardiomyocyte death (Figure 1F), suggesting a mechanism distinct from known cell death pathways in cardiomyocytes. Copper chelation with tetrathiomolybdate (TTM) rescued the cytotoxicity of the elesclomol–CuCl2 treatment. Elesclomol–CuCl_2_ treatment also significantly induced intracellular copper accumulation in AC16 cells (Figure 1G). The results indicated that elesclomol- CuCl2 induced cell death involved the intracellular copper accumulation. These results suggested that copper-induced cell death was indeed regulated, but was non-apoptotic, non-ferroptotic and non-necroptotic in cardiomyocytes.

This regulated programmed cell death induced by copper ions was previously termed cuproptosis in tumors and was characterized by a decrease in iron–sulfur (Fe–S) cluster proteins and a decrease in mitochondrial enzyme lipoylation. We next detected the hallmark changes of cuproptosis in cardiomyocytes after elesclomol–CuCl2 treatment. The expression of multiple mitochondrial Fe–S cluster proteins decreased, including FDX1, LIAS, ACO2, ETFDH and NDUFV1 (Figure 1I). To date, it is known that only four multimeric metabolic enzymes in mammals, including pyruvate dehydrogenase complexes (PDH), alpha-ketoglutarate complexes (KDH), branched-chain alpha-ketoacid dehydrogenase complex (BCKDH) and glycine cleavage system (GCV), could undergo posttranslational modification of lipoylation, yet these proteins are staples in the core metabolic landscape [30]. DLAT and DLST were the components E2 of PDH and KDH complexes, respectively. The lipoylation of DLAT and DLST is another hallmark of cuproptosis. We also found that the lipoylation of DLAT and DLST was decreased after elesclomol–copper treatment in cardiomyocytes (Figure 1H). Copper also decreased the expression of the mitochondrial respiratory chain complex (Figure 1J). Mitochondrial function, including intracellular ATP content (Figure 1K) and the mitochondrial complex I and III activity (Figure 1L), both decreased after elesclomol–CuCl2 incubation. These results indicated that mitochondrial function plays an important role in the copper-induced cell death of cardiomyocytes.

### 2.2. AGEs Induced Cuproptosis in Cardiomyocytes

The serum AGEs level of STZ-induced and db/db diabetic mice were both significantly elevated (Figure 2A). AGEs significantly induced the cell death of AC16, H9c2 and HL-1cardiomyocytes in a concentration-dependent manner (Figure 2B and Appendix A). AGEs, CuCl_2_ or elesclomol–CuCl2 promoted AC16 cell death alone, and the AGEs combined with CuCl_2_ or elesclomol–CuCl2 aggravated the cell death in AC16 cardiomyocytes (Figure 2C). The copper accumulation increased after AGEs treatment in cardiomyocytes, also suggesting that copper is involved the AGEs-induced cell death (Figure 2D). Treatment with multiple inhibitors of cell death mechanisms also suggested alleviation of AGE–CuCl_2_-induced cardiomyocyte death by TTM was more obvious than other inhibitors (Figure 2E).

Next, the mRNA expression of the copper import gene SLC31A1 was increased and the copper export genes ATP7A and ATP7B were decreased after AGEs treatment with or without CuCl_2_ (Figure 2F). Similarly, the protein expression of SLC31A1 was increased with the AGEs incubation time extending or with the AGEs concentrations increasing in AC16 and H9c2 cells (Figure 2G,H). These indicated that AGEs increased the SLC31A1 protein expression in cardiomyocytes in a concentration-dependent and time-dependent manner (Figure 2I,J). The Fe–S cluster proteins FDX1 and LIAS are the hallmarks of cuproptosis and were decreased after elesclomol–CuCl2 treatment. elesclomol–CuCl_2_ induced an increase in SLC31A1 and HSP70 expression in cardiomyocytes (Figure 2K). In particular, AGEs significantly promoted the changes in cuproptosis-related protein expression compared with the treatment of elesclomol–CuCl2 only (Figure 2K). Both the lipoic acid pathway and PDH complex are important pathways involved in cuproptosis. The RT-PCR analysis indicated the mRNA expression profile changes in the lipoic acid pathway after AGEs–CuCl_2_ treatment were similar to that of the elesclomol–CuCl2 treatment (Figure 2L). Meanwhile, the changes in the PDH complex pathway induced by AGEs–CuCl2 were also similar to those induced by elesclomol–CuCl2. These data indicated that AGEs could induce cuproptosis in cultured cardiomyocytes.

### 2.3. AGEs Regulated Copper Importer SLC31A1 Expression via ATF3/SPI1 in AC16 Cardiomyocytes

The ALGGEN PROMO, ChIP Atlas, Cistrome DB, GeneCards, GTRD, hTFtarget and UCSC JASPAR databases were integrated to predict possible transcription factors of SLC31A1. The upset-plot of the Venn diagram from the seven databases revealed a total of seven possible transcription factors: ATF2, ATF3, SPI1, ELF1, IRF1, RELA and HNF4A (Figure 3A). Next, we utilized the hTFtarget database to predict the possible co-transcription factors of the cuproptosis gene set including FDX1, LIAS, LIPT1, DLD, DLAT, PDHA1, PDHB, MTF1, GLS and CDKN2A. The intersection of the two predicted TF sets shows that ATF3 and SPI1 may play a role in the transcriptional regulation of both SLC31A1 and copper gene sets (Figure 3B).

Next, we validated the potential role of SPI1 and ATF3 in AC16 cardiomyocytes. The SPI1 and ATF3 mRNA expression increased ~20- or 30-fold after treatment with AGEs with or without CuCl_2_ in AC16 cardiomyocytes (Figure 3C). The changes in the other transcription factors, ATF2, ELF1, IRF1, RELA and HNF4A, were not as obvious as those in ATF3 and SPI1 (Figure 3C). In AC16 cells, AGEs also increased the expression of the two transcriptional factors ATF3 and SPI1 with the incubation time or concentrations increasing (Figure 3D,E). A time- and concentration-dependent increase in ATF3 and SPI1 protein expression was observed after AGEs treatment (Figure 3F,G). Meanwhile, compared with elesclomol–CuCl2 alone, the combination of AGEs remarkably promoted the SPI1 and ATF3 protein expression (Figure 3H). However, the protein expression increases in RELA and IRF1 were not significant compared with those in ATF3 and SPI1 (Figure 3H). Overexpression of ATF3 and/or SPI1 significantly increased the mRNA and protein expression of copper importer SLC31A1 in cardiomyocytes (Figure 3I,J).

### 2.4. AGEs/SPI1/ATF3/SLC31A1 Signaling Downregulated Mitochondrial Iron–Sulfur Cluster Protein Expression and Suppressed Mitochondrial Function

The previous study reported that iron–sulfur (Fe–S) proteins’ decline was the critical result of elesclomol–CuCl2 treatment [16]. The biogenesis of cellular iron–sulfur (Fe/S) proteins is the essential and minimal function of mitochondria [31]. This process is catalyzed by the bacteria-derived iron–sulfur cluster assembly (ISC) machinery and has been dissected into three major steps: de novo synthesis of a (2Fe-2S) cluster on a scaffold protein; Hsp70 chaperone-mediated trafficking of the cluster and insertion into (2Fe-2S) target apoproteins; and catalytic conversion of the (2Fe-2S) into a (4Fe-4S) cluster and subsequent insertion into recipient apoproteins [31]. ISC components of the first two steps terms as core ISC machinery are also required for biogenesis of numerous essential cytosolic and nuclear Fe/S proteins, explaining the essentiality of mitochondria [31]. The third step is also termed late ISC machinery, which is dominant for mitochondrial Fe/S proteins [31]. Hence, we detected the change in the Fe–S cluster after AGEs–CuCl2 treatment to explore whether the Fe–S cluster involved diabetic myocardial injury. RT-PCR was used to detect the mRNA changes of 20 mitochondrial Fe–S cluster genes involved in mitochondrial function after AGEs and AGEs–CuCl2 supplementation. Mitochondrial respiratory chain proteins NDUFS7, NDUFS8 and NDUFV1, as well as ACO2 and ETFDH involved in the tricarboxylic acid cycle and β-oxidation, were found to be significantly downregulated (Figure 4A). We also found that core ISC machinery proteins NFS1, ISD11 and FDXR, late ISC machinery proteins NFU1, BOLA3 and IND1, and transport protein ABCB7 were significantly decreased after AGEs and AGEs–CuCl2 treatment (Figure 4B).

LIAS is the regulator of lipoic acid metabolism and lipoylation; the lipoylation and oligomerization of mitochondrial proteins DLAT and DLST affect the function of mitochondria [16]. The previous study reported copper directly bound and promoted the oligomerization of lipoylated DLAT, which suppressed mitochondrial function [16]. Intriguingly, we also found that AGEs or AGEs–CuCl2 had a significant effect on lipoic acid metabolism in cardiomyocytes. Both expressions of DLAT, DLST, LIAS and the lipoylation of DLAT and DLST were decreased (Figure 4C). Meanwhile, the oligomerization of DLAT and DLST was induced by AGEs or AGEs–Cu (Figure 4C). AGEs and AGEs–CuCl2 similarly reduced the expression of the Fe–S cluster proteins NDUFS8, NDUFV1, ETFDH, ACO2, NFU1, NFS1 and ABCB7 (Figure 4D). Overexpression of ATF3 and SPI1 also played a similar role in reducing Fe–S cluster protein FDX1, ACO2 and DLST expression in cardiomyocytes (Figure 4E).

Western blotting analysis of mitochondrial oxidative respiratory chain complex also confirmed that AGEs inhibited mitochondrial respiration (Figure 4F). Meanwhile, AGEs and AGEs–CuCl2 reduced the ATP content (Figure 4G), as well as the activity of mitochondrial complexes I and III (Figure 4H,I) in cardiomyocytes. These results indicated that AGEs suppressed the mitochondrial function, and this was related to the decline of the Fe–S cluster.

### 2.5. Elevated Copper Levels in Peripheral Blood Circulation and Myocardial Tissue Aggravated Myocardial Damage in Diabetic Mice

Hearts were enlarged in both STZ-induced diabetic and db/db spontaneous diabetic mice (Figure 5A), and the heart weight (HW), body weight (BW) and heart weight/tibial length (HW/TL) ratio were larger than those of the control or db/bks mice (Figure 5B). Random blood glucose, fasting blood glucose and oral glucose tolerance test (OGTT) experiments all indicated the abnormal glucose levels and glucose tolerance in diabetic mice (Figure 5C,D). Transthoracic echocardiography also confirmed that the left ventricular ejection fraction (LVEF) and left ventricular fractional shortening (LVFS) were decreased, and the left ventricular end-systolic volume (LVESV) was significantly increased in diabetic mice (Figure 5E,F), indicating that the cardiac function of diabetic mice was decreased. Masson’s trichrome staining and Sirius Red staining showed that the degree of myocardial fibrosis was significantly increased in diabetic mice (Figure 5J).

The levels of copper ions in diabetic mice were significantly higher than those in the control group, both in serum and myocardial tissue (Figure 5G,H). Dithiooxamide staining was utilized to identify accumulated copper salts in tissue. We found that diabetes exacerbated the accumulation of copper salts in myocardial tissue (Figure 5I). These results indicated that copper might be involved in the process of myocardial injury in diabetes.

### 2.6. ATF3/SPI1/SLC31A1 Signaling of Cuproptosis Is Involved in Myocardial Injury in Diabetic Mice

ATF3, SPI1 and SLC31A1 were increased in diabetic myocardial tissue, indicating that the ATF3/SPI1/SLC31A1 signaling was activated (Figure 6A). We performed immunohistochemical staining for SLC31A1 and found increased expression in myocardial tissue (Figure 6B). The changes in the mRNA levels of lipoic acid pathway, PDH complex and Fe–S cluster genes in STZ and db/db diabetic mice were consistent with the cuproptosis change in cardiomyocytes in vitro (Figure 6C). We also observed a loss of Fe–S cluster proteins FDX1, LIAS, NDUFS8 and ACO2, as well as increased HSP70 abundance (Figure 6D) in the hearts of db/db mice. Meanwhile, the decrease in DLAT and DLST expression and decreased lipoylation of DLAT and DLAT were also significant in db/db mouse hearts (Figure 6D,E). The protein expression of mitochondrial oxidative respiratory chain complex in db/db murine hearts also declined compared with that in db/bks mice (Figure 6F). Meanwhile, the ATP content of heart tissue of db/db mice was decreased (Figure 6G). These findings in diabetic mouse models of copper toxicity suggested that copper overload resulted in the same cellular effects as those induced by AGEs combined with copper ionophores. Taken together, our data supported that excess copper promoted destabilization of Fe–S cluster proteins that resulted in mitochondrial dysfunction and, ultimately, cell death.

## 3. Discussion

Metal dyshomeostasis at various levels of an organism is a common denominator and cause or consequence of many illnesses [5]. Defective copper regulation mediates cardiovascular damage through two general processes that occur simultaneously in the same individual: elevation of Cu(II)-mediated pro-oxidant stress and impairment of copper-catalyzed antioxidant defense mechanisms [4,10]. Low concentrations of copper ions through coronary perfusion could significantly reduce cardiac function [11]. Excess copper was also proven to be mobilized following myocardial ischemia and facilitate tissue injury by catalyzing the production of hydroxyl radicals [32,33]. Long-term exposure to copper induces apoptosis in mouse hearts [34], but it is still unclear whether or how excess copper depresses cardiac function. Meanwhile, multiple lines of evidence have shown that the Cu(II)-selective chelators triethylenetetramine (TETA) and trientine could ameliorate left-ventricular hypertrophy and cardiac dysfunction in diabetic cardiomyopathy [12,13]. In this study, we found that copper in the serum and myocardium of STZ-induced and db/db spontaneous diabetic mice were both significantly increased. These results were consistent with those of previous studies [8,9,10,14]. This indicated that diabetes might facilitate copper overload, which could influence the cardiomyocyte function directly in diabetes. Diabetic cardiomyopathy is characterized by cardiomyocyte death, followed by cardiomyocyte hypertrophy, fibroblast proliferation and extracellular matrix increase, resulting in myocardial remodeling and cardiac dysfunction. However, the present limited study showed that copper aggravated diabetic cardiomyopathy mainly by increasing the extracellular matrix by activating TGF-beta/Smad signaling or oxidative stress [3]. It remaines elusive how copper directly influences the function of cardiomyocytes in diabetic cardiomyopathy [10,14,15], and there is no relevant study on the underpinning mechanism of copper overload-induced cuproptosis in diabetic cardiomyopathy.

Cuproptosis as the novel and unexpected programmed cell death triggered by targeting Cu to mitochondria has been demonstrated [16]. Copper ionophore elesclomol induced cuproptosis in a variety of tumor cells [16], but cuproptosis has not been investigated in cardiomyocytes. Our data suggest that copper overload also promotes cuproptosis in cardiomyocytes. We found elesclomol showed a higher affinity for copper than for other metal ions in cardiomyocytes. Compared with other copper ionophores, it exhibited stronger copper toxicity to cardiomyocytes. This copper-induced cuproptosis is also a unique form of irreversible damage and is only prevented by copper chelator TTM. Whole-genome and metabolism-focused CRISPR screens revealed that FDX1 functions as a direct binder of elesclomol. FDX1 donates an electron to lipoate synthase and reduces Cu^2+^ to Cu^+^, releasing it within the mitochondrial matrix [16,35]. Cuproptosis is elicited by the adventitious nonspecific binding of Cu to lipoylated proteins in mitochondria. FDX1 loss promotes accumulation and depletion of TCA cycle intermediates that correspond to specific reductions in metabolic enzyme lipoylation and accumulation of lipoylation DLAT [16]. We found similar characteristic changes in cardiomyocytes. Elesclomol–CuCl_2_ significantly induced a decrease in Fe–S cluster protein and decrease in lipoyl–DLAT and lipoyl–DLST and a significant abundance of HSP70 protein in cardiomyocytes. These factors contributed to decreased mitochondrial oxidative respiratory chain protein expression, depressed mitochondrial complex I and III activities and decreased ATP content. Depressed mitochondrial oxidative respiration is the main outcome of cuproptosis in cardiomyocytes, further suggesting the inability of the cells to respond to an energetic demand, which eventually leads to cardiomyocyte death.

AGEs are produced by glycation. AGEs are risk factors for multiple diabetes complications. Although the formation reaction of AGEs is very slow under physiological internal environment, this protein modification by glucose is significantly accelerated under the hyperglycemic conditions of diabetes [21,22,23]. The toxicity of AGEs on cardiomyocytes is mainly induced by calcium overload, oxidative stress or excessive autophagy [26,27,28]. Notably, catalytically-active Cu(II), probably binding AGEs and localizing themselves to blood vessels, contributes to pathogenic damage of atherosclerosis [10]. Meanwhile, a positive-feedback loop exists in Alzheimer’s disease: oxidative reactions catalyzed by Cu(II) are involved in the glycation of AGEs formation, and copper metalloproteins binding copper would undergo conformational changes and fragment after glycation, causing the release of catalytically-active Cu(II) [24,25]. However, the relationship between AGEs and cuproptosis has not been reported. We found that the concentrations of AGEs in the blood and myocardium were significantly upregulated in diabetic mice. Our data also illustrate that AGEs significantly induced cardiomyocyte death. More importantly, synergistic toxicity of AGEs and Cu was more obvious and could be alleviated by copper chelators to a certain extent, indicating that AGEs might promote copper overload-induced cardiotoxicity and participate in cuproptosis. We further analyzed the changes in copper transporters after AGEs treatment, and found that the mRNA and protein levels of copper importer SLC31A1 were significantly upregulated, while the exporters ATP7A and ATP7B were downregulated, indicating that AGEs might increase copper accumulation and thereby induce copper overload in cardiomyocytes. AGEs–CuCl2 similarly changed the cuproptosis-related protein expression and suppressed mitochondrial oxidative respiration. We also observed that SLC31A1 upregulation, cuproptosis-related protein expression changes and mitochondrial oxidative respiration decreases in diabetic murine hearts induced by STZ or db/db. To date, only the ATP7A protein is markedly downregulated in vessels isolated from T2DM patients and diabetic mice [36], whereas how the other copper transporters, ATP7B and SLC31A1, change in diabetes was uncertain. Our results indicated SLC31A1 was upregulated in DCM, which might be the reason for copper overload and cuproptosis in DCM. The upregulation of SLC31A1 might be an alternative illustration of the mechanism of DCM [16].

In addition, we explored the underlying mechanism of how AGEs upregulated the expression of SLC31A1 in AC16 cardiomyocytes. Multiple databases, includingALGGEN PROMO, ChIP Atlas, Cistrome DB, GeneCards, GTRD, hTFtarget and UCSC JASPAR, were integrated to predict possible transcription factors of SLC31A1. Meanwhile, the co-transcription factors of the cuproptosis gene set, including 10 genes of FDX1, LIAS, LIPT1, DLD, DLAT, PDHA1, PDHB, MTF1, GLS and CDKN2A, were predicted by the hTFtarget database. We found that ATF3 and SPI1 were the overlapping genes of the two predicted TF sets. The overexpression of ATF3 and/or SPI1 upregulated SLC31A1 and promoted the loss of Fe–S cluster protein expression. Our data indicated that ATF3 and/or SPI1 might be upstream of SLC31A1 and regulate cuproptosis of cardiomyocytes in diabetes (Figure 7). The transcriptional regulation of SLC31A1 by ATF3 and SPI1 has not been reported, not only in cardiovascular but also in diabetic disease. Activating transcription factor 3 (ATF3) is an adaptive-response gene. ATF3 plays a dual role in cardiac remodeling, and both ATF3 knockdown and overexpression in adult mice resulted in cardiac hypertrophy and dysfunction in response to pathological stimulation [37,38]. In diabetes, ATF3 also promoted β-cell dysfunction, accelerated STZ- induced diabetic liver injury and aggravated podocyte injury of db/db mice. This indicated the pathological effect of ATF3 in diabetes [39,40,41]. There is a positive feedback loop between ATF3 and oxidative stress [42,43]. Moreover, ATF3 could promote erastin-induced ferroptosis by repressing SLC7A11 expression and suppressing system Xc [44]. This indicated that ATF3 might be involved in pathogenicity by affecting redox homeostasis and regulating the expression of metal ion transporters. Our findings also proved that ATF3 might influence copper homeostasis and promote cuproptosis by regulating the expression of the copper importer SLC31A1. Spi-1 proto-oncogene (SPI1) is a multifaceted TF that involves numerous normal and pathogenic functions within the hematopoietic system [45]. However, there is less evidence investigating the relationship between SP1 and diabetes or cardiovascular diseases. A previous study has shown that SPI1 upregulation activated the TLR4/NFκB axis and aggravated myocardial infarction [46]. SPI1 also promoted the toxicity of nickel ions in THP-1 cells, indicating its important role in the homeostasis of metal ions [47]. Our findings revealed the potential regulatory effect of SPI1 on copper homeostasis and cuproptosis in diabetic cardiomyopathy.

This experiment indeed contained some limitations. Firstly, we performed most experiments in cell lines, but there still existed subtle differences between cell lines and primary cells. Hence, further investigation should be performed for the probable mechanism in primary cells. Next, further research is needed to determine whether intervention in the ATF3/SPI1/SLC31A1 signaling pathway could alleviate diabetic cardiovascular injury in diabetic animals. The exploration of inhibitors to related signaling pathways might also be a potential therapeutic target for the treatment of diabetic cardiomyopathy. Finally, ATF3/SPI1/SLC31A1 signaling might not be the only and the most important molecular target of AGEs–CuCl2 treatment. Whole-genome CRIPSR-Cas9 positive selection screen or Genome siRNA Library screen would find more underlying targets for AGEs–CuCl2 treatment.

In conclusion, our data showed, for the first time, that AGEs-induced cuproptosis might be a novel mechanism of DCM. The excessive increase in AGEs and copper in diabetes induced the upregulation of copper importer SLC31A1 through ATF3/SPI1, thereby mediating the accumulation of copper in cardiomyocytes, disturbing copper homeostasis and promoting cuproptosis. The decline in the Fe–S cluster protein and lipoylation of DLAT and DLST aggravated mitochondrial dysfunction in cardiomyocytes and resulted in myocardial dysfunction. Collectively, AGEs–CuCl2-induced cuproptosis via the ATF3/SPI1/SLC31A1 pathway might be an alternative potential therapeutic target for DCM.

## 4. Materials and Methods

### 4.1. Cell Culture and Compounds

The myocardial cell line H9c2 was obtained from the American Type Culture Collection. AC16 and HL-1 were donated by Hesong Zeng group of Division of Cardiology, Department of Internal Medicine, Tongji Hospital, Tongji Medical College and Huazhong University of Science and Technology (Wuhan, China). AC16 cardiomyocytes were validated by the STR profiling method. Cells were cultured in Dulbecco’s modified Eagle’s medium (DMEM, Keygen Biotech, Nanjing, China) supplemented with 10% (*v*/*v*) fetal bovine serum (FBS, Gibico, Thermo Fisher Scientific, Grand Island, NY, USA) and 1% (*v*/*v*) penicillin/streptomycin (Sangon, Shanghai, China) in a humidified atmosphere of 5% CO_2_ at 37 °C.

Metal ion chloride: Copper(II) chloride (CuCl_2_, 751944, Sigma-Aldrich, Darmstadt, Germany), cobalt(II) chloride (CoCl_2_, 232696, Sigma-Aldrich, Darmstadt, Germany), zinc chloride (ZnCl_2_, 208086, Sigma-Aldrich, Darmstadt, Germany), iron(III) chloride (FeCl_3_, 157740, Sigma-Aldrich, Darmstadt, Germany) and magnesium chloride (MgCl_2_, 208337, Sigma-Aldrich, Darmstadt, Germany) were purchased from Sigma-Aldrich Technology (Darmstadt, Germany). Nickel(II) chloride (NiCl_2_, N829833, Macklin, Shanghai, China) was purchased from Macklin Biochemical Technology (Shanghai, China). Iron(II) chloride (FeCl_2_, I106504, Aladdin, Shanghai, China) were purchased from Aladdin Biochemical Technology (Shanghai, China).

Copper ionophores: Elesclomol (S1052, Selleck, Shanghai, China) was purchased from Selleck (Shanghai, China). Thiram (45689, Sigma-Aldrich, Darmstadt, Germany), 2-Methyl-8-quinolinol (8-HQ, H57602, Sigma-Aldrich, Darmstadt, Germany) and tetramethylthiuram monosulfide (TMT, 567205, Sigma-Aldrich, Darmstadt, Germany) were purchased from Sigma-Aldrich. NSC-319726 (HY-18634, MCE, Shanghai, China), pyrithione-zinc (HY-B0572, MCE, Shanghai, China) and disulfiram (HY-B0240, MCE, Shanghai, China) were purchased from MedChemExpress Technology (MCE, Shanghai, China).

Cell death inhibitors: Ammonium tetrathiomolybdate (TTM, 323446, Sigma-Aldrich, Germany) was purchased from Sigma-Aldrich. L-NAME (HY-18729A, MCE, Shanghai, China), Boc-D-FMK (HY-13229, MCE, Shanghai, China) and Ferrostatin-1 (Fer-1, HY-100579, MCE, Shanghai, China) were purchased from MCE. Pepstatin A (S7381, Selleck, Shanghai, China), 3-Aminobenzamide (3-ABA, S1132, Selleck, Shanghai, China), N-Acetyl-L-cysteine (NAC, S1623, Selleck, Shanghai, China), necrostatin-1 (Nec-1, S8037, Selleck, Shanghai, China) and Z-VAD-FMK (S7023, Selleck, Shanghai, China) were purchased from Selleck.

Cell death inducers: Bortezomib (S1013, Selleck, Shanghai, China), Etoposide (S1225, Selleck, Shanghai, China) and ML162 (S4452, Selleck, Shanghai, China) were purchased from Selleck (Shanghai, China). Advanced glycation end products (AGEs, bs-1158P, bioss, Shanghai, China) were purchased from bioss Biochemical Technology (Shanghai, China).

### 4.2. Cell Viability Assay

Cellular viability was analyzed using the cell counting kit-8 (B34304, Selleck, Shanghai, China). The 5 × 10^3^ cells were seeded in 96-well plates and treated with indicated reagents for indicated time-points. A 10 μL CCK-8 solution was added to each well and the cultures were incubated at 37 °C for 45 min to 1 hour. Absorbance at 450 nm was measured using a microplate reader (Synergy 2, Bio-Tek Instruments, Winooski, VT, USA). The cell viability quantification in the heatmap was based on the absorbance at 450 nm. Four to six replicate wells for each group were established in every single experiment, and at least two independent times were carried out for each experiment. The absorbance of each group was normalized by the blank group (the indicated cells without any treatment). Given that the copper chelator TTM pretreatment significantly improved the cell viability, the absorbance of the TTM group was defined as 1, and the heatmap was plotted according to the average value of two independent experiments.

### 4.3. Caspase 3/7 Activity Analysis

Caspase 3/7 activation of caspase 3/7 cleavage in AC16 cells was detected by CellEvent Caspase-3/7 Green ReadyProbes (R37111, Invitrogen, Life Technologies Corporation, Eugene, OR, USA). AC16 cells were grown in a six-well in duplicate. Cells were subjected to elesclomol–CuCl2, Bortezomib and etoposide for 24 h and then incubated with caspase-3/7 green regent (2 drops/mL) for at least 30 min. The fluorescence image is detected through standard FITC filters by the MShot fluorescence microscope (Wuhan, China).

### 4.4. Intracellular Content of Copper

Intracellular content of copper was measured by copper assay kit (ab272528, Abcam, Boston, MA, USA). Then, 1 × 10^7^ AC16 cardiomyocytes after elesclomol–CuCl2, AGEs or AGEs–CuCl2 treatment were collected, and they were homogenized after adding 1 mL distilled water. Next, 100 µL samples per well were transferred to a flat-bottom 96-well UV plate. For each assay well, 35 µL Reagent, 5 µL Reagent B and 150 µL Reagent C were thoroughly mixed with samples, incubated for 5 min at room temperature and the optical density read at 359 nm using a microplate reader (Synergy 2, Bio-Tek Instruments, Winooski, VT, USA). Copper content was normalized by protein concentration.

### 4.5. ATP Content Detection

ATP content in cells and heart tissue was measured using a commercially available intracellular ATP assay kit (S0026, Beyotime, Shanghai, China). Previously, 100 μL ATP detection reagent was added to each well of 96-well plate and incubated at room temperature for 5 min to minimize the background. In addition, 20 μL cell or tissue lysate per well was then mixed with ATP detection reagent, and luminescence (RLU) was measured by microplate reader (Synergy 2, Bio-Tek Instruments, Winooski, VT, USA). ATP concentration was normalized by protein concentration.

### 4.6. Mitochondrial Fractionation

The mitochondrial extracts were isolated using the cell mitochondria isolation kit (C3601, Beyotime Biotechnology, Shanghai, China) according to the manufacturer’s protocol. Briefly, 5 × 10^7^ cells were washed with cold PBS twice. Ice-cold isolation reagent (800 μL, plus protease inhibitor) was added to the pellet and resuspended at 4 °C for 15 min. The suspension was homogenized about 10–30 times at 4 °C. After centrifuging at 600× *g* at 4 °C for 10 min, the supernatant was collected and transferred to a new Eppendorf tube. After centrifuging at 600× *g* at 4 °C for 10 min, the pellet of isolated mitochondria was ready for subsequent experiments.

### 4.7. Mitochondrial Complex I and III Activity

The extracted mitochondria of AC16 cardiomyocytes were used for mitochondrial complex activity measurement by CheKine micro mitochondrial complex I and III activity assay kits (KTB1850 and KTB1870, Abbkine, Wuhan, China). Mitochondria precipitation was resuspended by indicated reagents in steps. The samples and working reagents were mixed in a 96-well UV plate. The absorbance value was read at 0 min and 2 min after mixture and then was calculated according to the corresponding formula. The optimal absorbance of mitochondrial complex I and III was detected at 340 nm and 550 nm, respectively, by microplate reader (Synergy 2, Bio-Tek Instruments, Winooski, VT, USA).

### 4.8. Transcription Factor Prediction of Target Genes

ALGGEN PROMO (http://alggen.lsi.upc.es/ (accessed on 1 June 2022)), ChIP Atlas (https://chip-atlas.org/ (accessed on 1 June 2022)), Cistrome DB (http://cistrome.org/db/ (accessed on 1 June 2022)), GeneCards (https://www.genecards.org/ (accessed on 1 June 2022)), GTRD (http://gtrd.biouml.org/ (accessed on 1 June 2022)), hTFtarget (http://bioinfo.life.hust.edu.cn/hTFtarget/ (accessed on 1 June 2022)) and UCSC JASPAR (https://genome.ucsc.edu/ (accessed on 1 June 2022)) are seven databases used for predicting the possible transcription factors of target genes. We predicted the possible transcription factors of copper importer SLC31A1 (solute carrier family 31 member 1, Homo sapiens, Gene ID: 1317) online according to the detailed steps of each website. The cuproptosis gene set was obtained from the previously published article (PMID: 35298263). This gene set includes the seven genes of FDX1, LIAS, LIPT1, DLD, DLAT, PDHA1 and PDHB, positively associating with cuproptosis, and negatively cuproptosis-regulated genes of MTF1, GLS and CDKN2A. The co-transcription factors of the cuproptosis gene set were predicted by hTFtarget database. Overlapping TFs of the two predicted TF sets were identified by Venn diagram.

### 4.9. ATF3 and SPI1 Overexpression

Cells were transfected by seeding at a density of 100,000 cells/mL in 6-well plates in 2 mL of DMEM. After adherence, overexpression plasmid pcDNA3.1(+)-Flag-ATF3 and/or pcDNA3.1(+)-Flag-SPI1 (PAIWEI Technology, Wuhan, China) was transfected with Lipofectamine 3000 transfection reagent (L3000008, ThermoFisher) in Opti-MEM I reduced serum medium (31985070, ThermoFisher). After a 6-h incubation period, the medium was replaced with DMEM and cells were cultured for 48 h.

### 4.10. Whole Cell Protein Extraction and Western Blotting

Homogenate myocardial tissue and the collected cultured cells were lysed in RIPA lysis buffer with 1 mM protease inhibitor and phosphatase inhibitor for 40 min and then centrifuged at 12,000 rpm for 20 min at 4 °C. The supernatant was collected and quantified by BCA protein assay (P5026, Sangon, Shanghai, China). Proteins (15–30 μg) were loaded onto 8–15% Bis-Tris SDS-polyacrylamide gels and underwent electrophoresis at 60 V for 30 min and then 120 V for 1 hour. Then, separated proteins in gels were transferred to 0.45 μm PVDF membranes (Millipore, Tullagreen, Carrigtwohill, Ireland) at 230 mA for 100 min. After blocking with TBS-T (Tris-buffer saline with 0.1% Tween 20) containing 5% powdered milk for 90 min, the membranes were incubated with primary antibodies overnight. HRP-conjugated secondary antibodies and ultra-high sensitivity ECL (HY-K1005, MCE, Shanghai, China) were used for immunoblot imaging by ChemiScope 6100 Imager (QinXiang Products Ltd., Shanghai, China).

### 4.11. Primary Antibodies

PARP1 (#9532), cleaved caspase3 (#9664), VDAC (#4661), COX IV (#4844), ACO2 (#6571), DLST (#11954) and RELA (#8242) were obtained from Cell Signaling Technology (Beverly, MA, USA).

Lipoic acid (ab58724), total OXPHOS cocktail (ab110411) and FDX1 (ab108257) were obtained from Abcam Technology (Boston, MA, USA).

βTubulin (10068-1-AP), Vinculin (66305-1-Ig), LIAS (11577-1-AP) and GAPDH (10494-1-AP) were obtained from Proteintech Technology (Wuhan, China).

SLC31A1 (T510261), ATP7B (T58616) and HSP70 (M20033) were obtained from Abmart Technology (Shanghai, China).

ATF3 (A13469), SPI1 (A20461), IRF1 (A7692), DLAT (A8814), NDUFS8 (A13034), NDUFV1 (A8014), NFS1 (A13385), NFU1 (A7097) and ABCB7 (A14699) were obtained from ABclonal Technology (Wuhan, China).

### 4.12. Quantitative Real-Time Polymerase Chain Reaction (qRT-PCR)

Total RNA was isolated from frozen heart tissues or collected cultured cells using the FastPure Cell/Tissue Total RNA Isolation Kit (RC112, Vazyme, Nanjing, China). The RNA quality and concentration were determined spectrophotometrically (NanoDrop 2000 spectrophotometer, Thermo scientific, USA). Reverse transcription for cDNA synthesis was performed using HiScipt III RT SuperMix (R323, Vazyme, Nanjing, China). Quantitative real-time polymerase chain reaction was performed with SYBR green Fast qPCR mix (RK21203, ABclonal, Wuhan, China) on StepOnePlus (96-well, Life Technologies, Singapore) or LightCycler 480II (384-well, Roche Diagnostics, Switzerland) real-time PCR system. The mRNA expressions were normalized by housekeeping gene ACTB with the ΔΔCt method. The primer sequences are listed in the Appendix A.

### 4.13. Diabetic Mice Model

Male C57BL/6J, db/bks and db/db mice were purchased from the Charles River Laboratory Animal Technology Co., Ltd. (Beijing, China) and maintained on a regular chow diet. Two diabetic mice models were established: STZ-induced diabetic mice (*n* = 10) and control mice (*n* = 10); db/bks mice (*n* = 10) and db/db diabetic mice (*n* = 10). STZ-induced diabetes in C57BL/6J mice was performed by high-fat diet feeding (D12492, Research Diets, USA) for eight weeks and then an intraperitoneal injection of STZ (50 mg/kg body weight per day, S0130, Sigma-Aldrich Technology, Germany) in acetate phosphate buffer (C1013, pH 4.5, Solarbio, Beijing, China) for five consecutive days. Animals were considered diabetic when their blood glucose levels exceeded a pre-established value of 15 mmol/L (350 mg/dL). The db/db mice would show spontaneous elevated blood glucose and were considered diabetic when their blood glucose levels exceeded a pre-established value of 15 mmol/L (350 mg/dL) after 8 weeks. After 20 weeks of feeding, the mice were sacrificed. All animal experiments were supported by the institutional review board of Tongji Hospital, Tongji Medical College, Huazhong University of Science and Technology.

### 4.14. Blood Glucose and OGTT

Contour TS Blood Glucose Monitoring System (Ascensia Diabetes Care, Parsippany, NJ, USA) and blood glucose test strips (Ascensia Diabetes Care, USA) were used to measure blood glucose and oral glucose tolerance test (OGTT). To perform the OGTT, mice were fasted for 16 hours before receiving a gavage of 20% glucose (2 g/Kg) (G8150, Solarbio, Beijing, China), and blood samples were collected from the tail 0, 15, 30, 45, 60, 90 and 120 min later for blood glucose determination.

### 4.15. Serum AGEs Level

An Advanced Glycation End Products (AGEs) assay kit (ab273298, Abcam, USA) was used for detecting the AGEs’ serum levels. A 10 µL portion of serums and 190 µL AGEs assay buffer were added into wells of a white 96-well plate. The plate was then incubated at room temperature for 5 min and fluorescence (Ex/Em = 360/460 nm) measured at room temperature in end point mode using a fluorescence microplate reader (Synergy 2, Bio-Tek Instruments, Winooski, VT, USA). AGEs concentration was normalized by protein concentration.

### 4.16. Copper Levels in Serum and Myocardial Tissue

A copper colorimetric assay kit (E-BC-K300-M, Elabscience, Wuhan, China) was used for detecting copper levels in serum and myocardial tissue. Then, 20 µL of serum or heart tissue homogenate was added into wells of a 96-well plate, and 300 µL of detection reagent was mixed with samples. The plate was incubated at 37 °C for 5 min and optical density was read at 580 nm using a microplate reader (Synergy 2, Bio-Tek Instruments, USA). The copper concentration in heart tissue was normalized by protein concentration.

### 4.17. Transthoracic Echocardiography

Transthoracic echocardiography was performed before the mice were sacrificed (VINNO 6 VET, VINNO technology, Suzhou, China). Mice were anesthetized with isoflurane (3% for induction and 2% for maintenance) mixed in 1 L/min 100% O2 via a facemask. The left ventricular systolic function of the mice was evaluated by left ventricular ejection fraction (LVEF), fractional shortening (LVFS) and left ventricular end-systolic volume (LVESV). Cardiac parameters were measured and averaged from at least three separate cardiac cycles. The echocardiography operator was blinded to the grouping of mice.

### 4.18. Immunohistochemistry (IHC) Staining

A murine heart was fixed in a 10% formation fixative solution, conventionally dehydrated, embedded and the section was cut to 5μm thick. Tissue sections were deparaffinized and rehydrated through xylene and graded alcohols. Heat induced antigen retrieval was carried out with citrate buffer (pH = 6.0) and a pressure cooker at 122 °C for 45 s. Endogenous peroxidase activity was blocked by incubation, incubating sections for 15 min in 3% hydrogen peroxide and anhydrous ethanol (1:1). Sections were incubated at room temperature for 45 min with SLC31A1 antibody (1:500, T510261, Abmart). Application of the primary antibodies was followed by incubation for 30 min with HRP goat anti-rabbit IgG as a secondary antibody (ab6721, Abcam) and visualized with 3, 3′–diaminobenzidine (DAB) as a chromogen and hematoxylin counterstaining.

### 4.19. Cardiac Fibrosis Assessment

Cardiac fibrosis was evaluated by a Masson-trichrome staining kit (G1340, Solarbio, Beijing, China) and Sirius red staining kit (G1472, Solarbio, Beijing, China). Tissue sections were deparaffinized and rehydrated through xylene and graded alcohols. These sections were stained with Weigert’s Iron Hematoxylin Solution for 10 min and differentiated with Acid Alcohol Differentiation Solution for 10–15 s. Masson-trichrome staining was performed using the following steps: Blue in bluing Solution for 2–5 min, and rinse in deionized water. Then, stain with Ponceau-Acid Fucshin Solution for 10 min and rinse in deionized water. Differentiate in Phosphomolybic Acid Solution for 1–2 min or until collagen is not red. Without rinsing, add Aniline Blue Solution to the section and stain for 1–2 min. Finally, place sections in Acetic Acid Working Solution (Acetic Acid solution:deionized water = 1:2) for one minute. Sirius red staining was performed as follows: Sections were incubated for 30 min in Sirius Red Staining solution and subsequently washed with running water. The sections were then dehydrated in 95% ethanol, absolute ethanol and transparent xylene.

### 4.20. Copper Salt Staining

After conventionally dewaxing to water, heart tissue sections were stained with dithiooxamide staining solution (G3040, Solarbio Life Science, China) in a 37 °C water bath for 48 h. The section was rinsed with 70% ethanol and slightly washed with distilled water. Then, the section was dried and stained with a nuclear fast red solution for one minute. Next, it was washed with distilled water and conventionally dehydrated and transparently sealed with resinene. Copper salt was stained blackish green, and the nucleus was light red.

### 4.21. Reproducibility and Statistical Analysis

All experiments have been carried out at least two independent times. The indicated “n” in figure legends represents biological replicates. Data were expressed as the mean  ±  SD. A two-tailed student’s t-test or an ordinary one-way ANOVA followed by the Bonferroni post-test with multiple comparisons were used to compare the means among experimental groups and control.

## Figures and Tables

**Figure 1 ijms-24-01667-f001:**
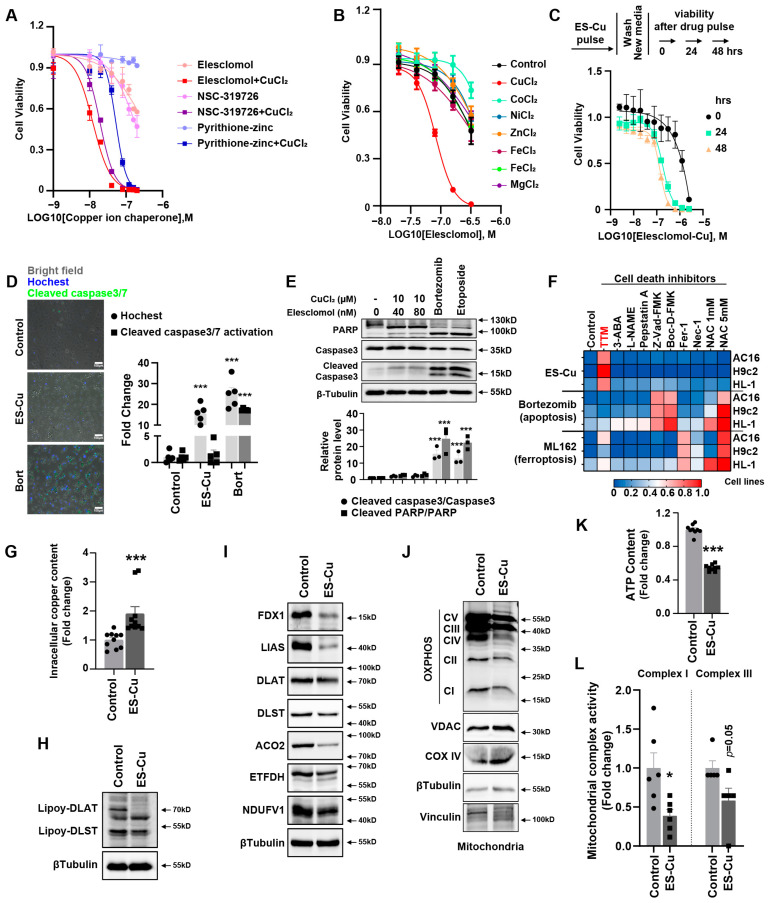
Cuproptosis is non-apoptotic, non-ferroptotic and non-necroptotic in cardiomyocytes. (**A**) Viability of AC16 cells after various treatments indicated copper ion chaperones with or without 10 μM CuCl_2_. (**B**) Viability of AC16 cells after treatment with elesclomol with or without 10 μM of indicated metals. (**C**) Viability of AC16 cells was assessed at the indicated times after elesclomol–Cu (1:1 ratio) pulse treatment and growth in fresh media. (**D**) Caspase 3/7 cleavage in AC16 cells after indicated treatments for 24 h (fold change over control). (**E**) Western blot analysis of AC16 cells treated with the indicated concentrations of elesclomol–Cu, 10 μM bortezomib or 10 μM etoposide for 24 h. (**F**) Heatmap of viability of AC16, H9c2 and HL-1 cardiomyocytes pretreated with 20 μM necrostatin-1, 10 μM ferrostatin-1, 1 mM N-acetylcysteine (NAC), 5 mM NAC, 30 μM Z-VAD-FMK, 50 μM D-Boc-FMK, 20 μM TTM, 300 μM L-NAME, 1 μM pepstatin A or 10 μM 3-ABA for 12 h and then treated with either elesclomol (40 nM)-CuCl_2_ (10 μM), 1 μM ML162 (GPX4 inhibitor) or 50 nM bortezomib for 48 h (average of three replicates). (**G**) Intracellular copper concentration in cardiomyocytes after ES–Cu treatment. (**H**) Western blot analysis of lipoylation of DLAT and DLST in AC16 cells treated with ES–Cu. (**I**) Western blot analysis of mitochondrial iron–sulfur cluster protein of AC16 cells treated with ES–Cu. (**J**) Western blot analysis of mitochondrial oxidative phosphorylation (OXPHOS) complex of AC16 cells treated with ES–Cu. (**K**) Intracellular ATP content of AC16 cells treated with ES–Cu. (**L**) Mitochondrial oxidative respiratory chain complex I and III activity of AC16 cells treated with ES–Cu. ES, elesclomol. * *p* < 0.05, *** *p* < 0.001.

**Figure 2 ijms-24-01667-f002:**
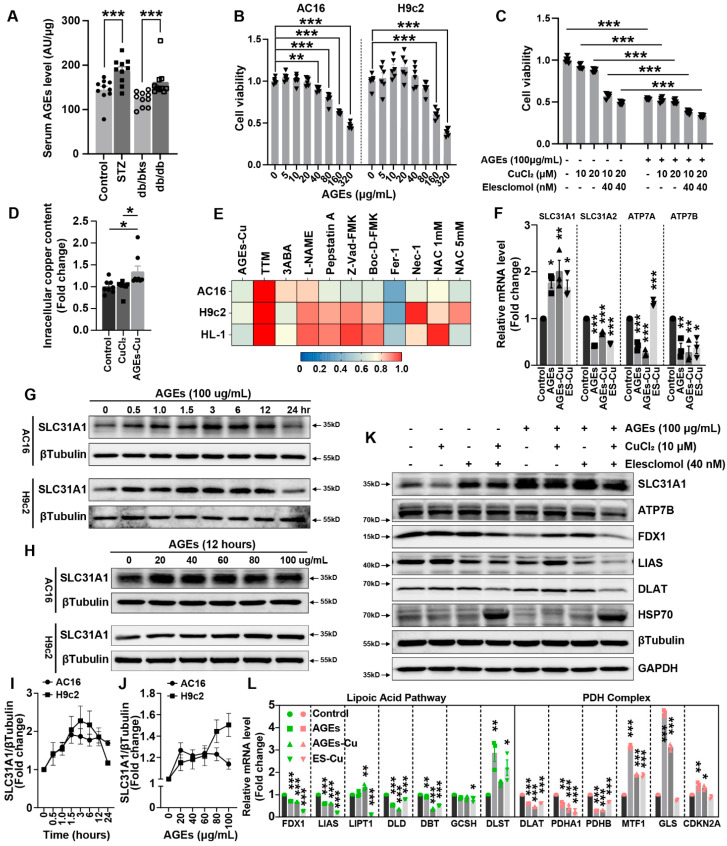
AGEs induce cuproptosis in cardiomyocytes. (**A**) Serum levels of AGEs in STZ-induced and db/db-diabetic mice. (**B**) Viability of AC16 and H9c2 cells after treatment with AGEs of indicated concentration. (**C**) Viability of AC16 cells after treatment with AGEs (100 μg/mL) and indicated elesclomol–Cu. (**D**) Intracellular copper content in cardiomyocytes after AGEs (100 μg/mL) and/or CuCl_2_ (10 μM) treatment. (**E**) Heatmap of viability of AC16, H9c2 and HL-1 cardiomyocytes pretreated with 20 μM necrostatin-1, 10 μM ferrostatin-1, 1mM N-acetylcysteine (NAC), 5 mM NAC, 30 μM Z-VAD-FMK, 50 μM D-Boc-FMK, 20 μM TTM, 300 μM L-NAME, 1 μM pepstatin A or 10 μM 3-ABA for 12 h and then treated with AGEs (100 μg/mL)-Cu (10 μM) for 48 h. (**F**) mRNA levels of copper ion transporter in AC16 cells treated with AGEs, AGEs–Cu or ES–Cu for 24 h. (**G**) Western blot analysis of SLC31A1 in AC16 or H9c2 cells treated with AGEs, with indicated time-points. (**H**) Western blot analysis of SLC31A1 in AC16 or H9c2 cells treated with AGEs with gradient concentrations. (**I**,**J**) The line charts of the quantifications of corresponding western blotting image in G and H. (**K**) Western blot analysis of cuproptosis-related protein in AC16 cells treated with indicated ES–Cu with and without AGEs. (**L**) mRNA levels of lipoic acid pathway and PDH complex in AC16 cells treated with AGEs, AGEs–Cu or ES–Cu for 24 h. ES, elesclomol. * *p* < 0.05, ** *p* < 0.01, *** *p* < 0.001.

**Figure 3 ijms-24-01667-f003:**
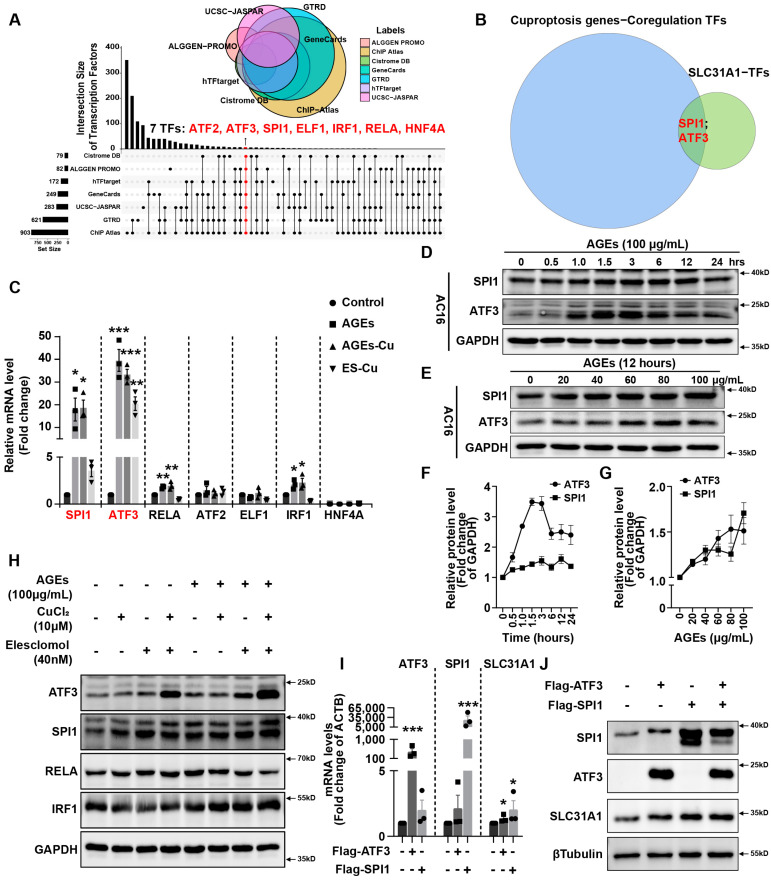
AGEs regulate SLC31A1 expression via ATF3/SPI1 in AC16 cardiomyocytes. (**A**) Upset-plot of Venn diagram of potential transcription factors of SLC31A1 predicted by varied databases. (**B**) Venn diagram of potential transcription factors between cuproptosis gene and SLC31A1. (**C**) mRNA levels of potential transcription factors in AC16 cells treated with AGEs, AGEs–Cu or ES–Cu for 24 h. (**D**) Western blot analysis of SPI1 and ATF3 of AC16 cells treated with AGEs with indicated time-points. (**E**) Western blot analysis of SPI1 and ATF3 of AC16 cells treated with AGEs with gradient concentrations. (**F**,**G**) The line charts of the quantifications of corresponding western blotting image in D and E. (**H**) Western blot analysis of SPI1 and ATF3 of AC16 cells treated with indicated ES–Cu with and without AGEs. (**I**) mRNA levels of SLC31A1 in AC16 cells overexpressed SPI1 or ATF3. (**J**) Western blot analysis of SLC31A1 in AC16 cells overexpressed SPI1 and/or ATF3. ES, elesclomol. * *p* < 0.05, ** *p* < 0.01, *** *p* < 0.001.

**Figure 4 ijms-24-01667-f004:**
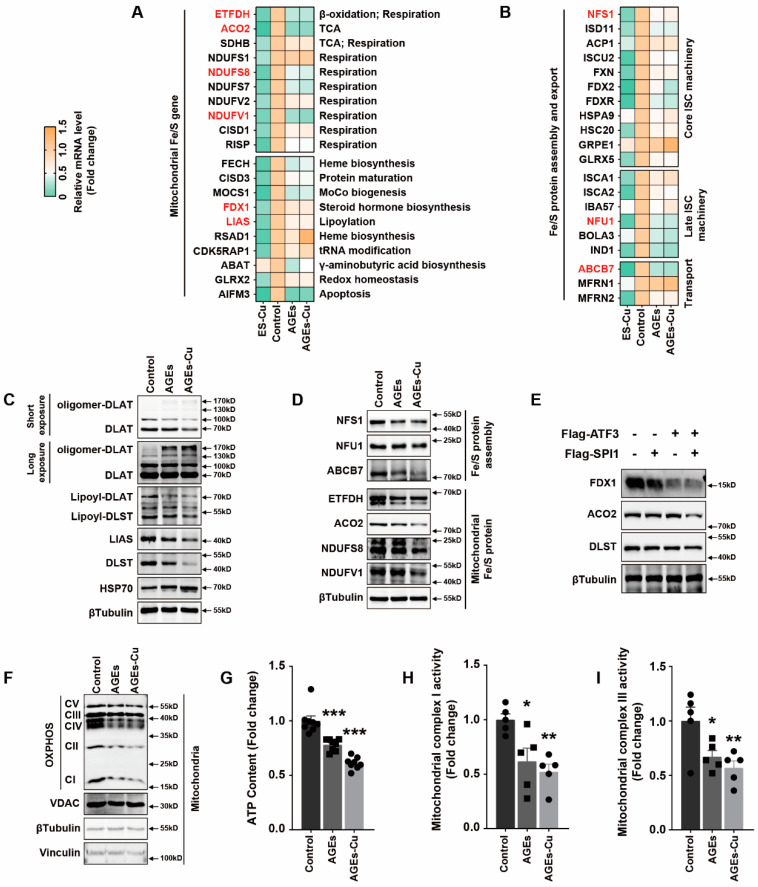
AGEs/ATF3/SPI1/SLC31A1 signaling decreased mitochondrial iron–sulfur cluster protein expression and mitochondrial function. (**A**,**B**) mRNA levels of mitochondrial iron–sulfur cluster (Fe–S) gene and Fe/S assembly and export in AC16 cells treated with AGEs, AGEs-Cu or ES–Cu for 24 h. (**C**) Western blot analysis of lipoylation and oligomerization of DLAT and DLST, LIAS, DLAT, DLST and HSP70 of AC16 cells treated with AGEs or AGEs–Cu. (**D**) Western blot analysis of iron–sulfur cluster protein in AC16 cells treated with AGEs or AGEs–Cu. (**E**) Western blot analysis of iron–sulfur cluster protein in AC16 cells overexpressed SPI1 and/or ATF3. (**F**) Western blot analysis of mitochondrial oxidative phosphorylation complex of AC16 cells treated with AGEs or AGEs–Cu. (**G**–**I**) Intracellular ATP content, mitochondrial oxidative respiratory chain complex I and III activity of AC16 cells treated with AGEs or AGEs–Cu. * *p* < 0.05, ** *p* < 0.01, *** *p* < 0.001.

**Figure 5 ijms-24-01667-f005:**
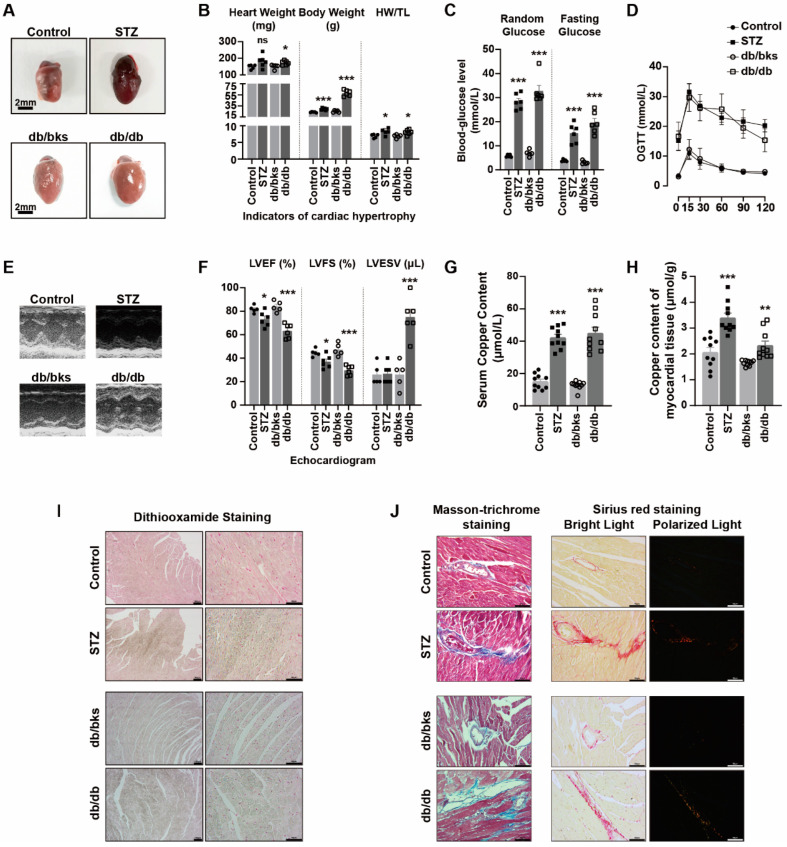
Elevated copper level of peripheral blood circulation and myocardial tissue aggravated myocardial damage in diabetic mice. (**A**) Representative gross morphologic pictures of murine heart. (**B**) Heart weight, body weight and HW/TL ratio of diabetic mice (*n* = 5–6). (**C**) Random and fasting glucose level of diabetic mice (*n* = 5–6). (**D**) Oral glucose tolerance test (OGTT) of diabetic mice (*n* = 5–6). (**E**) Representative M-mode echocardiographic changes in diabetic mice. (**F**) Left ventricular ejection fraction (LVEF), fractional shortening (LVFS) and left ventricular end-systolic volume (LVESV) of diabetic mice (*n* = 5–6). (**G**) Serum copper content of diabetic mice (*n* = 10). (**H**) Copper content of myocardial tissue in diabetic mice (*n* = 10). (**I**) Copper salt stained with dithiooxamide method in diabetic myocardial tissue by IHC. (**J**) Cardiac fibrosis with Masson-trichrome staining and Sirius red staining of diabetic mice. * *p* < 0.05, ** *p* < 0.01, *** *p* < 0.001.

**Figure 6 ijms-24-01667-f006:**
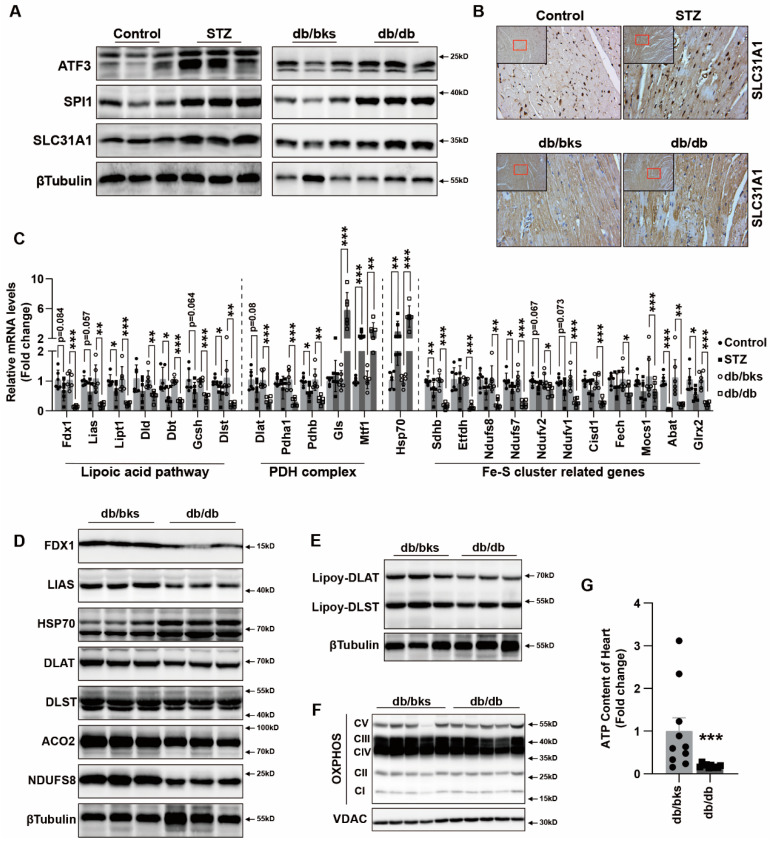
ATF3/SPI1/SLC31A1 signaling of cuproptosis involved in myocardium injury of diabetic mice. (**A**) Western blot analysis of ATF3/SPI1/SLC31A1 signaling of cuproptosis in myocardial tissue of diabetic mice (*n* = 3). (**B**) SLC31A1 expression of myocardial tissue in diabetic mice by IHC. (**C**) mRNA levels of cuproptosis and iron–sulfur cluster (Fe–S) gene in heart of STZ-induced and db/db- diabetic mice (*n* = 6). (**D**) Western blot analysis of Fe–S cluster proteins in myocardial tissue of diabetic mice (*n* = 3). (**E**) Western blot analysis of lipoylation of DLAT and DLST in myocardial tissue of diabetic mice (*n* = 3). (**F**) Western blot analysis of the mitochondrial oxidative phosphorylation complex in myocardial tissue of diabetic mice (*n* = 5). (**G**) ATP content in myocardial tissue of diabetic mice (*n* = 10). * *p* < 0.05, ** *p* < 0.01, *** *p* < 0.001.

**Figure 7 ijms-24-01667-f007:**
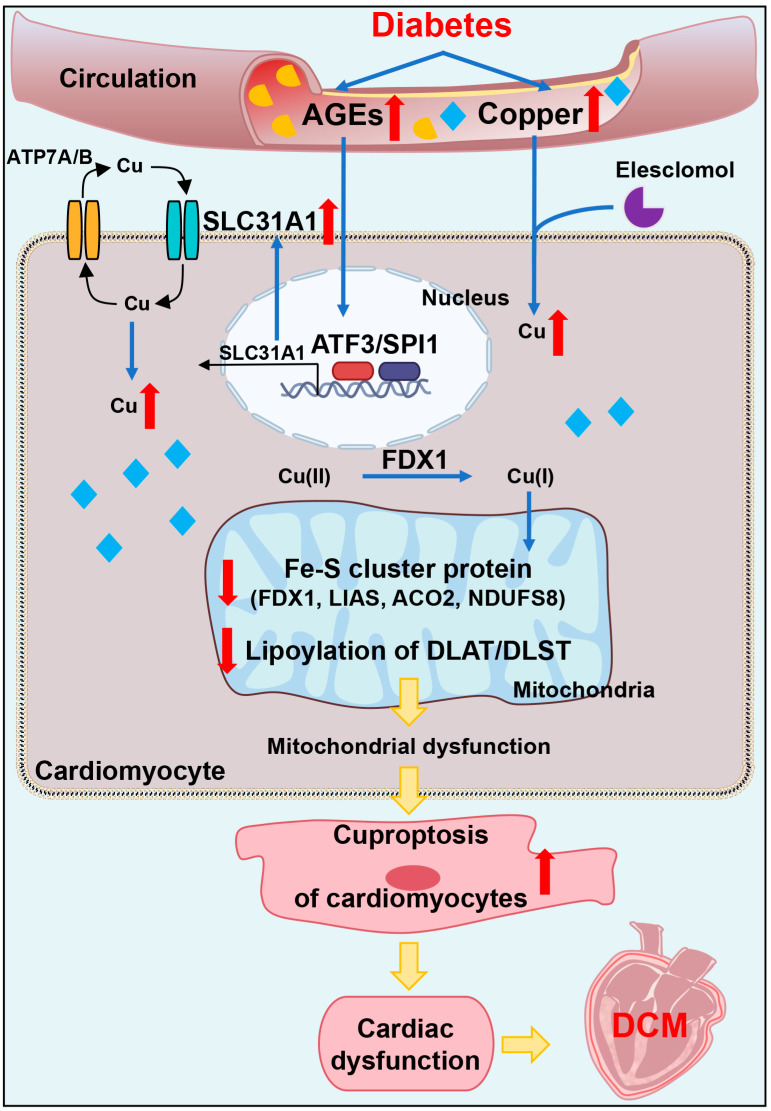
Schematic of mechanisms of AGEs–CuCl2-induced cuproptosis via ATF3/SPI1/SLC31A1 pathway in diabetic cardiomyopathy. Excessive AGEs and copper in diabetes upregulated copper importer SLC31A1 through ATF3/SPI1, thereby mediating copper accumulation in cardiomyocytes, disturbing copper homeostasis and promoting cuproptosis. This promoted the decline of Fe–S cluster protein (FDX1, LIAS, NDUFS8 and ACO2) and decreased lipoylation of DLAT- and DLST-aggravated mitochondrial dysfunction in cardiomyocytes and resulted in myocardial dysfunction.

## Data Availability

All data generated or analyzed during this study are included in this published article and its Appendix A.

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
