# Peer review of "ATF3/SPI1/SLC31A1 Signaling Promotes Cuproptosis Induced by Advanced Glycosylation End Products in Diabetic Myocardial Injury"

_ijms, 2023, doi:10.3390/ijms24021667_

Round 1

Reviewer 1 Report

In this study Shengqi Huo et al. investigated the mechanism by which copper (Cu) overload caused by persistent hyperglycemia in diabetes can trigger mitochondrial dysfunction, cell death and ultimately aggravate diabetic cardiomyopathy (DCM). Specifically, the authors have shown that the pathological increase of advanced glycosylation end products (AGEs) in diabetes induces the upregulation of copper importer SLC31A1 through ATF3/SPI1, thereby mediating the accumulation of copper in cardiomyocytes, disturbing copper homeostasis and promoting cuproptosis. This is an innovative study that highlights the link between AGEs-induced cuproptosis and DCM. The experimental design is well written and the type of experiments have been properly chosen to support the main claims of the paper. However, in some cases statistical analysis are missing and only descriptive sentences are given. 

I have a few comments that will further increase the clearness of the message of the study: 

 Figure 1A: The graph is extremely crowded, it is difficult to discern between a specific ionophore alone or in combination with copper chloride. Please make the image more readable.

 The authors stated that (line 101) : “Two-hour pulse treatment with elesclomol-copper chloride triggered cell death that lasted for 48 hours (Figure 1C)” Cell death is a no turning back point. When cells die after elesclomol-copper chloride pulse, cells just die, and claiming that cell death lasted for 48 hours makes no sense. Please rephrase it.

Figure 2E: Could the authors specify the methodology used for cell viability quantification in the heatmap? Is it based on absorbance at 450 nm?

 Figure 2B: Sometimes 3 different cell lines are used to strengthen the results, but this approach is not consistent in the manuscript (e.g. in Figure 2B HL-1 cell line is missing). Please comment.

The authors stated that (136-137) “Moreover, AGEs further aggravated copper-induced AC16 cardiomyocyte death with or without copper ionophore-elesclomol treatment (Figure 2C). This indicated AGEs might induce cell death in the copper-dependent manner.” This statement is not supported by data at this point of the manuscript. It could be merely an additive effect without a direct causative correlation. Please rephrase.

 Figure 2G-H. The authors stated that: “Similarly, AGEs increased the SLC31A1 protein expression in cardiomyocytes in a concentration-dependent and time-dependent manner." There is no quantification for these western blotting (here and in other panels). Without quantification and numbers is difficult to assert that. Please add quantification(s).

 Typos:

line 30 “AGEs triggered cardiomyocyte death and aggravated when”

line 46 “metabolic diseases including obesity and diabetes (are) usually accompanied by dysfunctional regulation of varied metal ions”

line 96 “To future confirm copper ionophore cytotoxicity”ine

line 145: the word “obviously” is inappropriately used in the manuscript.

Line 189: West(ern) blotting analysis

Author Response

Point 1: Figure 1A: The graph is extremely crowded, it is difficult to discern between a specific ionophore alone or in combination with copper chloride. Please make the image more readable.

Response 1: Thanks for your professional suggestion. We reorganized Figure 1A, three copper ionophores including elesclomol, NSC-319726, and Pyrithione-zinc, significantly reduced the AC16 cell viability when combined with copper chloride. More importantly, elesclomol-CuCl2 showed the most pronounced cell-killing effect on AC16 cells. The cytotoxicity of other copper ionophores, like thiram, disulfiram, 8HQ, and TMT, was not obvious compared with the three ionophores above in AC16 cells. Hence, these data were shown in Supplementary Figure 1.

Point 2: The authors stated that (line 101): “Two-hour pulse treatment with elesclomol-copper chloride triggered cell death that lasted for 48 hours (Figure 1C)” Cell death is a no turning back point. When cells die after elesclomol-copper chloride pulse, cells just die, and claiming that cell death lasted for 48 hours makes no sense. Please rephrase it.

Response 2: We appreciate for your careful comment, our expression about Figure 1C is somewhat perplexing. We rephrased the representation in the manuscript. Two-hour pulse treatment with elesclomol-copper chloride triggered cell death significantly. Meanwhile, the cells were treated with two-hour pulse elesclomol-copper chloride (ES-Cu) and then washed with a new medium to remove the ES-Cu, these cells were cultured for an additional 24 hours or 48 hours, and the CCK8 assay revealed that the cell viability was decreased with continued culture. These results indicated the cell death was irreversible.

Point 3: Could the authors specify the methodology used for cell viability quantification in the heatmap? Is it based on absorbance at 450 nm?

Response 3: Thanks for your detailed comment on improving the accessibility of our manuscript. We indeed used the CCK8 assay to evaluate the cell viability. The cell viability quantification in the heatmap was based on the absorbance at 450 nm. 4-6 replicate wells for each group were established in every single experiment, and at least two independent times were carried out for each experiment. The absorbance of each group was normalized by the blank group (the indicated cells without any treatment). Given the copper chelator TTM pretreatment significantly improved the cell viability, the absorbance of TTM group was defined as 1, the heatmap was plotted according to the average value of two independent experiments. We have revised the method of cell viability detection in the manuscript.

Point 4: Figure 2B: Sometimes 3 different cell lines are used to strengthen the results, but this approach is not consistent in the manuscript (e.g. in Figure 2B HL-1 cell line is missing). Please comment.

Response 4: Thanks for your professional comment. The use of multiple cell lines to validate relevant results is a critical approach to improve the reliability of the experimental evidence. We did use three different cell lines, AC16, H9c2, and HL-1 to verify the reproducibility of research results. In particular, we proved that the cell death induced by elesclomol-CuCl2 could be rescued by copper chelator TTM (Figure 1F and 2E) in three lines of cardiomyocytes and AGEs could induce cell death in AC16 and H9c2 cells (Figure 2B). Therefore, we added Supplementary Figure 2, showing that in HL-1 myocardial cells, AGEs could also significantly reduce cell viability, which is consistent with the conclusions of the other two cell lines. For subsequent experiments related to intracellular mechanisms, we mainly performed experiments in AC16 cells, because the three cell lines came from three different species, the reagents, kits or primary antibodies required for some experiments were not responsive to all species, so we selected human AC16 cells for all mechanism verification.

Point 5: The authors stated that (136-137) “Moreover, AGEs further aggravated copper-induced AC16 cardiomyocyte death with or without copper ionophore-elesclomol treatment (Figure 2C). This indicated AGEs might induce cell death in the copper-dependent manner.” This statement is not supported by data at this point of the manuscript. It could be merely an additive effect without a direct causative correlation. Please rephrase.

Response 5: We appreciate your careful comment. Our statement might be biased, so we rephrased the representation. AGEs, CuCl2 or elesclomol-CuCl2 promoted AC16 cell death alone, and the AGEs combined with CuCl2 or elesclomol-CuCl2 aggravated the cell death in AC16 cardiomyocytes (Figure 2C). AGEs combined with CuCl2 also induced intracellular copper increase compared with a single CuCl2 treatment (Figure 2D). This indicated copper might involve AGEs-related cell death. We also revised the related representation in this manuscript.

Point 6: Figure 2G-H. The authors stated that: “Similarly, AGEs increased the SLC31A1 protein expression in cardiomyocytes in a concentration-dependent and time-dependent manner." There is no quantification for these western blotting (here and in other panels). Without quantification and numbers is difficult to assert that. Please add quantification(s).

Response 6: Thank you very much for the professional suggestion. We quantified the western blotting image of Figure 2G-2H and Figure 3D-3E. The line charts of the quantifications were placed next to the corresponding western blotting image. The results indicated that the protein expression of SLC31A1 was increased with the AGEs incubation time extending or with the AGEs concentrations increasing in AC16 and H9c2 cells (Figure 2I, 2J). In AC16 cells, AGEs also increased the expression of the two transcriptional factors ATF3 and SPI1 with the incubation time or concentrations increasing (Figure 3F, 3G). We revised the result part of the manuscript.

Point 7: Typos: Line 30 “AGEs triggered cardiomyocyte death and aggravated when”. Line 46 “metabolic diseases including obesity and diabetes (are) usually accompanied by dysfunctional regulation of varied metal ions”. Line 96 “To future confirm copper ionophore cytotoxicity”. Line 145: the word “obviously” is inappropriately used in the manuscript. Line 189: West(ern) blotting analysis

Response 7: Thank you very much for the detailed review. We have sent the manuscript for English polishing and carefully proofread the manuscript to correct all the grammar and typos.

Reviewer 2 Report

This interesting manuscript described the Cuproptosis in diabetic hearts and AGEs-treated cells. The experiments were well designed and data were solid. There are only some small concerns:

1. The English needs to be improved.

2. The source of cells should be clear. In addition, the company information of some regents should follow the reqirement (City, state, country).

Author Response

Point 1: The English needs to be improved.

Response 1: Thanks for your professional comments on improving the accessibility of our manuscript. We have sent the manuscript for English polishing and carefully proofread the manuscript to correct all the grammar and typos.

Point 2: The source of cells should be clear. In addition, the company information of some regents should follow the reqirement (City, state, country).

Response 2: Thank you very much for the detailed review. We have added the detailed information about relevant reagents.

Reviewer 3 Report

In this study, the Authors investigated whether copper overload affected cuproptosis and if it was involved in AGEs-induced cardiotoxicity and ultimately in the prevention of diabetic cardiomyopathy. A diabetic (streptozotocin-induced ) murine model and AC16 cardiomyocytes were used to test the hypothesis . The Authors' findings might be insightful  for the clinical arena in the context of diabetic cardiomyopathy.

The work has merit, it is narrowly conducted, and well structured.

The design is sound although a power analysis (why 10 animals per group) would have been of assistance.

On a singular note, the introduction would benefit from citing similar papers on such model/purposes:

  • 10.1016/j.dld.2019.09.002
  • 10.5650/jos.ess18086
  • Please, in the introduction, avoid contracted form like "it's" instead of "it is" (line 87)

Author Response

Point 1:

The design is sound although a power analysis (why 10 animals per group) would have been of assistance.

Response 1: Thank you for your advice. The animal experiments in this research have been approved by the institutional animal care and use committee of Huazhong University of Science and Technology (approval code: 2953). Ethical approval contained the sample size calculation. We used the Power & Animal Number Calculator tool provided by the Centre for Comparative Medicine Research, The University of Hong Kong. According to the estimated result, the number of animals needed in each control and each test group at 90% power and at p<0.05 between each group was eight. Hence, we designed ten animals per group in this experiment.

Point 2:

On a singular note, the introduction would benefit from citing similar papers on such model/purposes:

10.1016/j.dld.2019.09.002

10.5650/jos.ess18086

Response 2: Thank you for your kind suggestion. We revised part of our introduction to enrich the representation more comprehensive.

Point 3:

Please, in the introduction, avoid contracted form like "it's" instead of "it is" (line 87)

Response 3: Thank you for your kind suggestion. We have revised the improper representation and carefully check the whole manuscript.

Reviewer 4 Report

In the manuscript titled “ATF3/SPI1/SLC31A1 Signaling Promotes Cuproptosis Induced by Advanced Glycosylation End Products in Diabetic Myocardial Injury”, their study showed that excessive AGEs and copper in diabetes upregulated ATF3/SPI1/ SLC31A1 signaling, thereby disturbing copper homeostasis and promoting cuproptosis for the first time. It would be an alternative potential therapeutic target for DCM. This is a relatively interesting discovery. I have the following suggestions to improve the manuscript:

• In the part of “Results”, the authors referred that “The results indicated that elesclomol-induced cell death involved the intracellular copper accumulation but not the effect of the small molecule chaperones themselves.”, the evidence for the author's statement is not sufficient. It would be better if citations or more convincing figure of cell viability evidence could be provided. You can just put the groups you want to show in figure 1A, other groups can be put in supplement figures. Figure 1B has the same problem.

• In the part of “Caspase 3/7 activity analysis”, the authors referred that “Elesclomol-CuCl2 induced cell death without the caspase 3/7 cleavage, which is significantly different from the cell death induced by bortezomib. Bortezomib resulted in a 20- to 30-fold increase in levels of caspase 3/7 activation.”, it would be better if the authors could explain what’s the difference between them and elaborate on what the staining means and how the comparisons are made.

• In the part of “Results”, the authors referred that “Similarly, AGEs increased the SLC31A1 protein expression in cardiomyocytes in a concentration-dependent and time-dependent manner (Figure 2G, 2H).”, The author only provides a representative figure. It is not convincing, and the representative figure does not conform to “a concentration-dependent and time-dependent manner”. It would be better for the authors to provide bar graphs and explain their results more accurately.

Regarding the result part of Figure 4, it would be better if the authors could make a more detailed description of this process. This explanation is so confusing. It is recommended that the author refer to more literature.

• In the part of “Discussion”, authors are advised to also discuss the difficulties encountered in the research and the limitations of their study.

• Please check the typo in the manuscript, for example,” To future (further) confirm copper ionophore cytotoxicity is selectively dependent on copper,”.

Author Response

Point 1: In the part of “Results”, the authors referred that “The results indicated that elesclomol-induced cell death involved the intracellular copper accumulation but not the effect of the small molecule chaperones themselves.”, the evidence for the author's statement is not sufficient. It would be better if citations or more convincing figure of cell viability evidence could be provided. You can just put the groups you want to show in figure 1A, other groups can be put in supplement figures. Figure 1B has the same problem.

Response 1: Thanks for your professional suggestion. We reorganized Figure 1A, three copper ionophores including elesclomol, NSC-319726, and Pyrithione-zinc, significantly reduced the AC16 cell viability when combined with copper chloride. More importantly, elesclomol-CuCl2 showed the most pronounced cell-killing effect on AC16 cells. The cytotoxicity of other copper ionophores, like thiram, disulfiram, 8HQ, and TMT, was not obvious compared with the three ionophores above in AC16 cells. Hence, these data were shown in Supplementary Figure 1. The Figure 1B was also modified for the clear demonstration.

We also rechecked the representation “The results indicated that elesclomol-induced cell death involved the intracellular copper accumulation but not the effect of the small molecule chaperones themselves.” This expression was indeed improper to interpret Figure 1A. Combined with the result of Figure 1G, the elesclomol-CuCl2 significantly induced the copper accumulation in AC cells. We move the representation to the behind of Figure 1G, and rephrased it to “The results indicated that elesclomol- CuCl2 induced cell death involved the intracellular copper accumulation.”. Previous reports also revealed that elesclomol assists copper shuttling from the extracellular to the intracellular compartments, leading to continued copper accumulation within mitochondria1. Elesclomol and its analogs induced oxidative stress and correlated with the elevation of mitochondrial copper levels and cytotoxic activity1, 2. These also illustrated copper accumulation played an important role in the effect of elesclomol. We also carefully checked the manuscript and revised it.

Point 2: In the part of “Caspase 3/7 activity analysis”, the authors referred that “Elesclomol-CuCl2 induced cell death without the caspase 3/7 cleavage, which is significantly different from the cell death induced by bortezomib. Bortezomib resulted in a 20- to 30-fold increase in levels of caspase 3/7 activation.”, it would be better if the authors could explain what’s the difference between them and elaborate on what the staining means and how the comparisons are made.

Response 2: We appreciate your careful comment. We used caspase 3/7 positive cells to evaluate the apoptotic cell level. The caspase 3/7 positive cells were green fluorescent. The number of total cells at white bright image was calculated by Image J software. The quantification of the caspase3/7 activity was based on the number of green fluorescent positive cells to the total cell numbers. Bortezomib could target multiple pathways including p53, the nuclear factor kappa B, the phosphatidylinositol 3 kinase pathway, and the ubiquitin/proteasome pathway to promote apoptosis3. The Figure 1D revealed that bortezomib induced green-positive apoptotic cells increase. However, the number of green-positive apoptotic cells after elesclomol-CuCl2 treatment was not increased. This indicated elesclomol-CuCl2 induced cell death was not related to caspase 3/7 activation. We also revised the manuscript.

Point 3: In the part of “Results”, the authors referred that “Similarly, AGEs increased the SLC31A1 protein expression in cardiomyocytes in a concentration-dependent and time-dependent manner (Figure 2G, 2H).”, The author only provides a representative figure. It is not convincing, and the representative figure does not conform to “a concentration-dependent and time-dependent manner”. It would be better for the authors to provide bar graphs and explain their results more accurately.

Response 3: Thank you very much for the professional suggestion. We quantified the western blotting image of Figure 2G-2H and Figure 3D-3E. The line charts of the quantifications were placed next to the corresponding western blotting image. The results indicated that the protein expression of SLC31A1 was increased with the AGEs incubation time extending or with the AGEs concentrations increasing in AC16 and H9c2 cells (Figure 2I, 2J). In AC16 cells, AGEs also increased the expression of the two transcriptional factors ATF3 and SPI1 with the incubation time or concentrations increasing (Figure 3F, 3G). We revised the result part of the manuscript.

Point 4: Regarding the result part of Figure 4, it would be better if the authors could make a more detailed description of this process. This explanation is so confusing. It is recommended that the author refer to more literature.

Response 4: Thanks for your professional comments on improving the accessibility of our manuscript. We supplied more interpretation for Figure 4, and revised the manuscript. The previous study reported that iron-sulfur (Fe-S) proteins decline was the critical result of elesclomol-CuCl2 treatment4. Fe-S cluster protein is an important participant in the normal biological function of mitochondria. The biogenesis of cellular iron-sulfur (Fe/S) proteins is the essential and minimal function of mitochondria5. This process is catalyzed by the bacteria-derived iron-sulfur cluster assembly (ISC) machinery and has been dissected into three major steps: de novo synthesis of a [2Fe-2S] cluster on a scaffold protein; Hsp70 chaperone-mediated trafficking of the cluster and insertion into [2Fe-2S] target apoproteins; and catalytic conversion of the [2Fe-2S] into a [4Fe-4S] cluster and subsequent insertion into recipient apoproteins5. ISC components of the first two steps terms as core ISC machinery are also required for biogenesis of numerous essential cytosolic and nuclear Fe/S proteins, explaining the essentiality of mitochondria5. The third step is also termed late ISC machinery, which is dominant for mitochondrial Fe/S proteins5. Hence, we detected the change of Fe-S cluster after AGEs- CuCl2 treatment to explore whether Fe-S cluster involved diabetic myocardial injury. RT-PCR was used to detect the mRNA changes of 20 mitochondrial Fe-S cluster genes involved in mitochondrial function after AGEs and AGEs-CuCl2 supplementation. Mitochondrial respiratory chain proteins NDUFS7, NDUFS8 and NDUFV1, as well as ACO2 and ETFDH involved in tricarboxylic acid cycle and β-oxidation, were found to be significantly downregulated (Figure 4A). The biosynthesis of iron-sulfur cluster proteins includes the assembly and transport. We also found that core ISC machinery proteins NFS1, ISD11 and FDXR, late ISC machinery proteins NFU1, BOLA3 and IND1, and transport protein ABCB7 were significantly decreased after AGEs and AGEs-CuCl2 treatment (Figure 4B).

LIAS is the regulator of lipoic acid metabolism and lipoylation, the lipoylation and oligomerization of mitochondrial proteins DLAT and DLST affect the function of mitochondria4. The previous study reported copper directly bound and promoted the oligomerization of lipoylated DLAT, which suppressed mitochondrial function4. Intriguingly, we also found that AGEs or AGEs-CuCl2 had a significant effect on lipoic acid metabolism in cardiomyocytes. Both expressions of DLAT, DLST, LIAS and the lipoylation of DLAT and DLST were obviously decreased (Figure 4C). Meanwhile, the oligomerization of DLAT and DLST was induced by AGEs or AG-Es-Cu (Figure 4C). AGEs and AGEs-CuCl2 similarly reduced Fe-S cluster protein NDUFS8, NDUFV1, ETFDH, ACO2, NFU1, NFS1 and ABCB7 expression (Figure 4D).

We would like to thank you again for your professional comment, which has further improved the professionalism and readability of our manuscript.

Point 5: In the part of “Discussion”, authors are advised to also discuss the difficulties encountered in the research and the limitations of their study.

Response 5: Thank you for your kind suggestion. discuss the difficulties encountered in the research and the limitations of their study. This experiment indeed contained some limitations. Firstly, we performed most experiments in cell lines, but there still existed subtle differences between cell lines and primary cells. Hence, further investigation should be performed for the probable mechanism in primary cells. Next, further research is needed to determine whether intervention in ATF3/SPI1/SLC31A1 signaling pathway could alleviate diabetic cardiovascular injury in the diabetic animals. The exploration of inhibitors to related signaling pathways might also be a potential therapeutic target for the treatment of diabetic cardiomyopathy. Finally, ATF3/SPI1/SLC31A1 signaling might not be the only and the most important molecular target of AGEs-CuCl2 treatment. Whole-genome CRIPSR-Cas9 positive selection screen or Genome siRNA Library screen would find more underlying targets for AGEs-CuCl2 treatment.

Point 6: Please check the typo in the manuscript, for example,” To future (further) confirm copper ionophore cytotoxicity is selectively dependent on copper,”.

Response 6: Thank you for your kind suggestion. We have sent the manuscript for English polishing, revised the improper representation and carefully check the whole manuscript.

References

  1. Nagai M, Vo NH, Shin Ogawa L, Chimmanamada D, Inoue T, Chu J, et al. The oncology drug elesclomol selectively transports copper to the mitochondria to induce oxidative stress in cancer cells. Free Radic Biol Med. 2012;52(10):2142-50.
  2. Hasinoff BB, Yadav AA, Patel D, Wu X. The cytotoxicity of the anticancer drug elesclomol is due to oxidative stress indirectly mediated through its complex with Cu(II). J Inorg Biochem. 2014;137:22-30.
  3. Ghobrial IM, Witzig TE, Adjei AA. Targeting apoptosis pathways in cancer therapy. CA Cancer J Clin. 2005;55(3):178-94.
  4. Tsvetkov P, Coy S, Petrova B, Dreishpoon M, Verma A, Abdusamad M, et al. Copper induces cell death by targeting lipoylated TCA cycle proteins. Science. 2022;375(6586):1254-61.
  5. Lill R, Freibert SA. Mechanisms of Mitochondrial Iron-Sulfur Protein Biogenesis. Annu Rev Biochem. 2020;89:471-99.
